

# Longevity and survival of *Leptocybe invasa* (Hymenoptera: Eulophidae), an invasive gall inducer on *Eucalyptus*, with different diets and temperatures

Amanda Rodrigues De Souza[1,*], Leonardo Rodrigues Barbosa[2,*], José Raimundo de Souza Passos[3,*], Bárbara Monteiro de Castro e Castro[4,*], José Cola Zanuncio[5,*] and Carlos Frederico Wilcken[1,*]

[1] Faculdade de Ciências Agronômicas, Departamento de Proteção Vegetal, UNESP (São Paulo State University), Botucatu, São Paulo, Brazil
[2] Embrapa Florestas, Empresa Brasileira de Pesquisa Agropecuária, Colombo, Paraná, Brazil
[3] Instituto de Biociências, Departamento de Bioestatística, UNESP (São Paulo State University), Botucatu, São Paulo, Brazil
[4] Departamento de Fitotecnia, Universidade Federal de Viçosa, Viçosa, Minas Gerais, Brazil
[5] Departamento de Entomologia/ BIOAGRO, Universidade Federal de Viçosa, Viçosa, Minas Gerais, Brazil
[*] These authors contributed equally to this work.

Corresponding author
Bárbara Monteiro de Castro e Castro, barbara.monteiro@ufv.br, barbaram-castro@hotmail.com

## ABSTRACT

The blue gum chalcid, *Leptocybe invasa* Fisher & LaSalle (Hymenoptera: Eulophidae), causes galls on *Eucalyptus* spp. leaf midribs, petioles and stems. Biological aspects need to be studied to assist in developing management strategies and to maintain this insect in the laboratory to rear the parasitoid *Selitrichodes neseri* Kelly & La Salle (Hymenoptera: Eulophidae) that depends on having a supply of *Eucalyptus* seedlings infested by *L. invasa*. We evaluated the longevity and survival of *L. invasa* individual non reproductive females fed with six different diets (pure honey, 50% honey solution, pure honey plus eucalyptus leaves, eucalyptus leaves, distilled water, or no food) and seven different temperatures (10, 14, 18, 22, 26, 30 and 34 °C). *Leptocybe invasa* fed with 50% honey solution and reared at 14 or 18 °C showed the greatest longevity and survival.

## INTRODUCTION

The blue gum chalcid, *Leptocybe invasa* Fisher & La Salle (Hymenoptera: Eulophidae) is a native Australian wasp and an important *Eucalyptus* spp. pest worldwide (*Mendel et al., 2004*). This insect was first reported in Mediterranean and Middle Eastern regions in 2000 (*Mendel et al., 2004*) and later in 39 countries (*Nugnes et al., 2015*). In Brazil, *L. invasa* was found for the first time on *Eucalyptus grandis* × *Eucalyptus camaldulensis* hybrid clones in northeastern Bahia in 2007 (*Costa et al., 2008*; *Wilcken et al., 2015*). This invasive wasp inserts its eggs in the epidermis of young plant parts, such as leaf midribs, petiole and stems (*Zhu, Qiu & Ren, 2013*; *Wilcken et al., 2015*), causing galls. These galls deform leaves and

terminal shoots, resulting in defoliation and drying of shoots, thereby reducing seedling and tree growth (*Wilcken et al., 2015*).

Highly efficient control strategies to prevent damage and manage *L. invasa* are unknown (*Zheng et al., 2014*; *Wilcken et al., 2015*). Chemical control showed varying success (*Nyeko, Mutitu & Day, 2007*; *Basavana Goud et al., 2010*; *Kulkarni, 2010*; *Jhala, Patel & Vaghela, 2010*; *Javaregowda, Prabhu & Patil, 2010*). The high costs of chemical products and their likely negative effect on beneficial insects make this method an unfeasible option at a large plantation scale. Biological control using parasitoids constitute one *L. invasa* management strategy (*Kim et al., 2008*; *Gupta & Poorani, 2009*; *Dittrich-Schröder et al., 2012*). The parasitoid *Selitrichodes neseri* Kelly and La Salle (Hymenoptera: Eulophidae) has potential for the biological control of this pest (*Kelly et al., 2012*) due to its relatively short developmental time, long adult life span when supplemented with carbohydrates, ability to utilize a range of gall ages and its high host specificity (*Dittrich-Schröder et al., 2014*). Successful release and establishment of the parasitoid *S. neseri* in a *Eucalyptus* plantation in Brazil was demonstrated (*Masson et al., 2017*). *Selitrichodes neseri* can obtain parasitism rates over 70% in *L. invasa* populations in laboratory (*Dittrich-Schröder et al., 2014*). However, the *S. neseri* mass rearing depends on having a supply of *Eucalyptus* seedlings infested by *L. invasa*.

Many insects, such as the parasitoids *Lysibia nana* Gravenhorst and *Gelis agilis* Fabricius (Hymenoptera: Ichneumonidae) and *C leruchoides noackae* Lin and Huber (Hymenoptera: Mymaridae), depend on supplementary nutritional sources, such as sugar and carbohydrates, to maintain their metabolism and increase their survival (*Harvey et al., 2012*; *Souza et al., 2016*). Food supplements (*Schmale et al., 2001*; *Hossain & Haque, 2015*), including pollen, nectar and honeydew can increase the longevity (*Lee, Heimpel & Leibee, 2004*), fecundity and flight capacity (*Winkler et al., 2009*), besides increasing hymenopteran survival (*Luo et al., 2010*).

Temperature can also affect insect survival (*Burgi & Mills, 2013*; *Colinet et al., 2015*; *Zhu et al., 2015*), embryonic development, behavior and reproduction (*Liu et al., 2015*). Extreme temperatures can reduce organism fitness (*Singh, Kochar & Prasad, 2015*) and holometabolic insects such as *L. invasa* can suffer stress at different life cycle stages. Thermal stress, such as high temperatures (*Lieshout, Tomkins & Simmons, 2013*; *Zizzari & Ellers, 2011*), can affect longevity, fecundity and fertility (*Hance et al., 2007*; *Nguyen, Bressac & Chevrier, 2013*), and low temperatures can reduce survival, fecundity, reproduction (*Lacoume, Bressac & Chevrier, 2007*; *Singh, Kochar & Prasad, 2015*) and mobility (*Ayvaz et al., 2008*) of insects.

The *L. invasa* life-cycle varies according to the eucalypt species and climatic conditions. Mean survival period for wasps fed with honey and water was 6.5 days and for three days without food at 25 °C and developmental time from oviposition to emergence was 132.6 days (*Mendel et al., 2004*). Honey solution could prolong the longevity of *L. invasa* females (*Sangtongpraow, Charernson & Siripatanadilok, 2011*). Biological characteristics, such as reproduction, fecundity, oviposition behavior (*Sangtongpraow, Charernson & Siripatanadilok, 2011*), host range (*Mendel et al., 2004*) survival and longevity of *L. invasa* fed on different diets and exposed to different temperatures, need to be studied to

 

understand the relationship between population expansion and the environment factors aimed at optimizing the rearing of this pest in the laboratory. The dispersion, methods of field survey and adult biology of *L. invasa* have been studied (*Tang et al., 2008*; *Wu et al., 2009*; *Zhu et al., 2011*; *Zhu et al., 2012*), but temperature thresholds and diets that may affect survival and longevity of this insect require further research. This information is essential to maintaining colonies of this insect in the laboratory that are essential to the rearing of the parasitoid *S. neseri*. Therefore, the aim of this study was to evaluate the effects of different diets and temperatures on longevity and survival of *L. invasa* individual non reproductive females.

## MATERIALS AND METHODS

*Leptocybe invasa* females were obtained from a colony reared on *E. grandis × E. camaldulensis* seedlings at the Laboratory of Biological Control of Forest Pests (LCBPF) of the School of Agricultural Sciences, in Botucatu, São Paulo State, Brazil (22°50′48,14″S; 48°25′53,52″W; 786 m). The insects are reared in cages with pure honey under in a climate room at $25 \pm 2$ °C, $70 \pm 10\%$ RH with photoperiod of 12:12 h L:D in *Eucalyptus* seedlings.

### Diet bioassays

Newly-emerged *L. invasa* females (virgins) were placed individually in glass vials (2.5 cm diameter × 8.5 cm height) capped with plastic film and fed with different diets: pure honey (100%) (T1), 50% honey solution (T2), pure honey plus eucalyptus leaves (T3), eucalyptus leaves (T4), distilled water (T5), or no food (T6). Honey was chosen as food source by ease of use in laboratory rearing and eucalyptus leaves provide excellent food sources for *L. invasa*, helping to build up abundant populations in field (*Jacob & Ramesh, 2009*). Food was replaced every two days. Wasps were held in a climate chamber at $25 \pm 2$ °C, $70 \pm 10\%$ RH with photoperiod of 12:12 h L:D. *Leptocybe invasa* survival was evaluated daily.

### Temperatures bioassays

Newly-emerged *L. invasa* females (virgins) were placed individually in glass vials (2.5 cm diameter × 8.5 cm height) capped with plastic film and maintained at temperatures of 10, 14, 18, 22, 26, 30 and 34 °C in a climate chamber at $70 \pm 10\%$ RH with photoperiod of 12:12 h L:D. The temperature of 26 °C was used because it is standard in rearing laboratories for many insects, and other temperatures (14, 18, 22 and 30 °C) are representative of field conditions in Brazil. The temperatures of 10 and 34 °C were chosen as extreme conditions. These wasps were fed with a 50% honey solution, with food replaced every two days.

The experimental design was completely randomly with 25 replications. Therefore, 150 individuals were used for the diet experiment and 175 individuals for the temperature experiment. *Leptocybe invasa* survival was evaluated daily.

### Statistical analysis

Survival curves for *L. invasa* females were analyzed using Kaplan–Meier product-limit estimator (*Lee, 1992*) using SAS (SAS *University edition*; SAS, Inc., Cary, NC, USA). These were compared using the Log-Rank test adjusted by Sidak ($P < 0.05$) (*Westfall et al., 1999*),

**Table 1** Adult longevity (days) and range (days) (mean ± SE) of *Leptocybe invasa* (Hymenoptera: Eulophidae) females fed with different diets at 25 ± 2 °C, 70 ± 10% RH and photoperiod of 12:12 h L:D (*n* = 25).

| Treatments | Longevity | Range |
|---|---|---|
| Pure honey (100%) | 7.08 ± 2.54 a | 2–12 |
| Honey 50% | 8.92 ± 2.75 a | 2–12 |
| Honey 100% + Eucal. leaves | 7.00 ± 2.91 a | 3–12 |
| Eucal. leaves | 2.16 ± 0.89 b | 1–5 |
| Distilled water | 1.44 ± 0.50 b | 1–2 |
| Control (no food) | 1.56 ± 0.50 b | 1–2 |

**Notes.**
Means followed by the same letter do not differ by the Sidak test ($P < 0.05$).

according to two bioassays: the first experiment considered six diets and the second seven temperatures.

## RESULTS

Diet affected *L. invasa* female longevity. Longevity of adult wasps was greatest when fed with 50% honey solution, 100% honey and 100% honey plus eucalyptus leaves, respectively. Although there is no statistical difference for longevity of wasps fed on these three diets, there seems to be some biological difference. Wasps fed a 50% honey solution lived longer. Adding foliage to the honey treatment did not change longevity compared to the wasps fed on honey alone. Wasps fed only eucalyptus leaves showed similar longevity with those that were fed only with water or had no food provided (Table 1). Similarly, among the three groups of females that did not receive honey, there were no significant differences in survival. Females which received eucalyptus leaves as food survived much longer than those receiving water or no food (Fig. 1).

With regard to the influence of temperature on survival and longevity, longevity of *L. invasa* females was highest at 14 °C (33.28 ± 3.18 days) and 18 °C (29.36 ± 3.82 days) and lower at temperatures of 10 °C and above 22 °C (Table 2). Wasp survival was highest at 14 °C and 18 °C. Females exposed to temperatures of 10 °C and above 22 °C showed lower survival (Fig. 2).

## DISCUSSION

Diet and temperatures affected *L. invasa* female longevity and survival. Wasp fed on the 50% honey solution, 100% honey and 100% honey plus eucalyptus leaves and reared in 14 °C and 18 °C fed on the 50% honey solution presented greater survival and longevity.

The observation of increased *L. invasa* female longevity for wasps provided with honey solution is similar to that found for other Hymenoptera under laboratory conditions (*Harvey et al., 2012*; *Hossain & Haque, 2015*; *Souza et al., 2016*). For example, *Cleruchoides noackae* Lin and Huber (Hymenoptera: Mymaridae) survived 1.2 days without food, and 3.4, 3.3 and 3.7 days when provided with honey solution at 100%, 50%, and 10%, respectively (*Souza et al., 2016*). *Microplitis croceipes* Cresson (Hymenoptera: Braconidae)
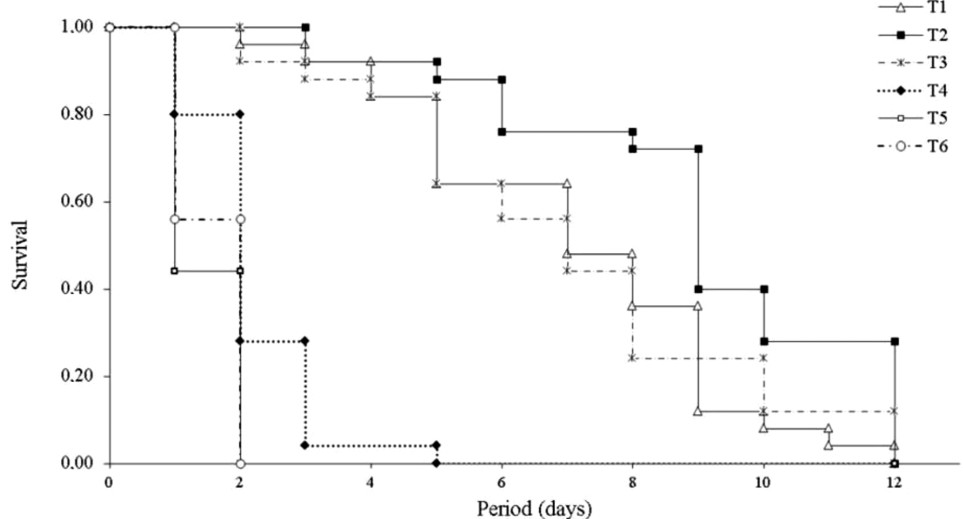

**Figure 1** **Survival (days) of *Leptocybe invasa* (Hymenoptera: Eulophidae) females reared with different diets.** Survival (days) of *Leptocybe invasa* (Hymenoptera: Eulophidae) females reared with 100% honey (T1), 50% honey solution (T2), 100% honey supplemented with eucalyptus leaves (T3), eucalyptus leaves (T4), distilled water (T5) and no food (T6) at 25 ± 2 °C, 70 ± 10% RH and photoperiod of 12:12 h L:D ($n = 25$).

**Table 2** **Adult longevity (days) and range (days) (mean ± SE) of *Leptocybe invasa* (Hymenoptera: Eulophidae) females exposed to different temperatures (°C) at 70 ± 10% RH and photoperiod of 12:12 h L:D ($n = 25$).**

| Temperature | Longevity (days) | Range |
|---|---|---|
| 10 °C | 11.48 ± 2.80 bc | 6–15 |
| 14 °C | 33.28 ± 3.18 a | 25–40 |
| 18 °C | 29.36 ± 3.82 ab | 16–35 |
| 22 °C | 7.76 ± 1.36 cd | 5–9 |
| 26 °C | 7.56 ± 1.47 cd | 2–9 |
| 30 °C | 1.96 ± 0.78 d | 1–3 |
| 34 °C | 1.72 ± 0.73 d | 1–3 |

**Notes.**
Means followed by the same letter do not differ by the Sidak test ($P < 0.05$).

females longevity was found to be four times greater when provided with honey compared to those receiving water only (*Nafziger Jr & Fadamiro, 2011*), while *Dinarmus basalis* (Rondani) (Hymenoptera: Pteromalidae) females survived 27.9 and 25.3 days when provided with honey and sugar solution at 10%, respectively, but only 19.7 days with distilled water only (*Hossain & Haque, 2015*). A supply of honey was also found to extend the longevity of *Ophelimus eucalypti* (Hymenoptera: Eulophidae) (*Withers, Raman & Berry, 2000*). In the present study, *L. invasa* longevity when not provided with honey was similar to that found for *Trichogramma maxacalli* Voegelé and Pointel (Hymenoptera: Trichogrammatidae), where adults were found to survive longer on diets containing sugars than without food (*Oliveira et al., 2003*). These observations can be explained by the honey

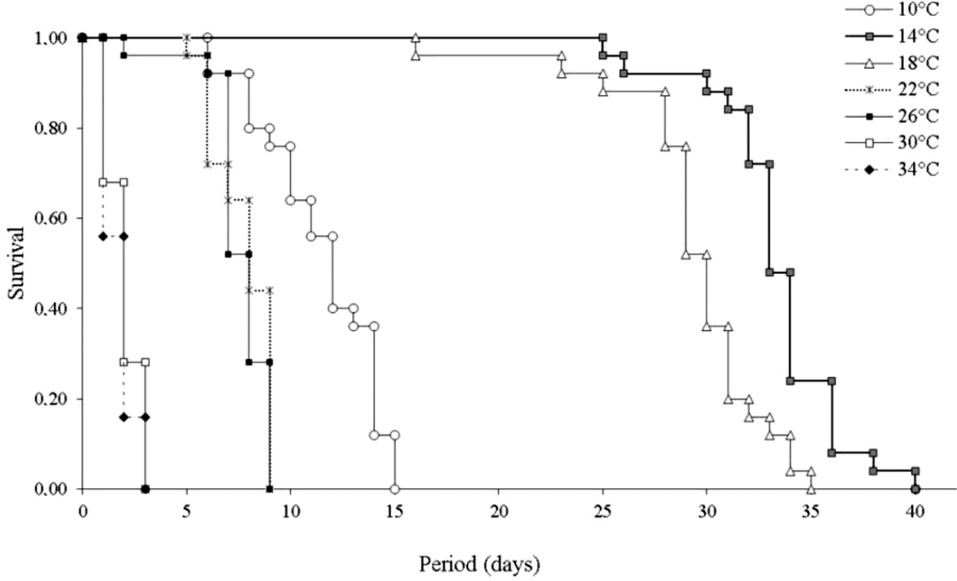

**Figure 2  Survival (days) of *Leptocybe invasa* (Hymenoptera: Eulophidae) females at different temperatures.** Survival (days) of *Leptocybe invasa* (Hymenoptera: Eulophidae) females fed with a 50% honey solution at different temperatures, 70 ± 10% RH and photoperiod of 12:12 h L:D (*n* = 25).

constitution, with this food providing at least 181 nutrients such as amino acids, enzymes, minerals, proteins, sugars and vitamins (*Alvarez-Suarez et al., 2009*).

*Leptocybe invasa* longevity and survival were affected by temperature, but in the field this insect thrives under a wide temperature range (*Mendel et al., 2004*). Invasive species can adapt to environmental factors, developing stronger tolerance and wider niche breadth. Tolerance to temperature and other factors co-determined invasive species distribution and opportunity for future regional dispersion (*Agrawal, 2001*; *Bale, Masters & Hodkinson, 2002*; *Bale & Hayward, 2010*). Differences in *L. invasa* longevity in relation to temperature agree with reports for other hymenopterans such as *Meteorus ictericus* (Nees) (Hymenoptera: Braconidae) (*Burgi & Mills, 2013*) and *C. noackae* (*Souza et al., 2016*). *Leptocybe invasa* longevity recorded at low temperatures in this study was similar to that of *Trichospilus diatraeae* Cherian and Margabandhu (Hymenoptera: Eulophidae) (*Rodrigues et al., 2013*).

Low *L. invasa* adult survival at higher temperatures was similar to that observed for *Microplitis manilae* (Hymenoptera: Braconidae) (*Qiu et al., 2012*). Increased survival at low temperatures can be associated with reduced activity and metabolism (*Bleicher & Parra, 1990*) and low females survival at higher temperatures may be due to metabolic increases or enzyme destruction at higher temperatures (*Mohan, Verma & Singh, 1992*). On the other hand, these results are inconsistent with the distribution of this pest in tropical and subtropical regions, which may be explained by microclimatic effects with a lower impact due to variable temperatures in the field compared to the constant temperatures to which wasps were exposed to in the laboratory (*Zhu et al., 2015*). The wasp behavior to

avoiding high temperatures, such as seeking shade or cooling with wing fluttering and the population and/or strain origin can also aid in adapting to different conditions.

## CONCLUSION

Provision of honey as a food and temperatures of 14 and 18 °C increase *L. invasa* longevity and survival. These results are important to understanding this pest biology, which is essential to enhance methods of mass-rearing *L. invasa* and make feasible rearing of parasitoids in laboratory. Studies should aim to explore resource utilization efficiencies in adult parasitoids reared on different diets and temperatures focusing on their reproductive strategies.

## ACKNOWLEDGEMENTS

Dr. Phillip Villani revised and corrected the English language used in this manuscript.

### Funding

This work was financially supported by ''Conselho Nacional de Desenvolvimento Cientifico e Tecnológico (CNPq) (Process: 142131/2012-1), Fundação de Amparo a Pesquisa do Estado de Minas Gerais (FAPEMIG)'' and ''Programa Cooperativo sobre Proteção Florestal (PROTEF)'' of the ''Instituto de Pesquisas e Estudos Florestais (IPEF)''. The funders had no role in study design, data collection and analysis, decision to publish, or preparation of the manuscript.

### Grant Disclosures

The following grant information was disclosed by the authors:
Conselho Nacional de Desenvolvimento Cientifico e Tecnológico (CNPq): Process: 142131/2012-1.
Fundação de Amparo a Pesquisa do Estado de Minas Gerais (FAPEMIG).
Instituto de Pesquisas e Estudos Florestais (IPEF).

### Competing Interests

The authors declare there are no competing interests.

### Author Contributions

- Amanda Rodrigues De Souza conceived and designed the experiments, performed the experiments, analyzed the data, contributed reagents/materials/analysis tools, prepared figures and/or tables, authored or reviewed drafts of the paper, approved the final draft.
- Leonardo Rodrigues Barbosa conceived and designed the experiments, performed the experiments, authored or reviewed drafts of the paper, approved the final draft.
- José Raimundo de Souza Passos analyzed the data, authored or reviewed drafts of the paper, approved the final draft.

- Bárbara Monteiro de Castro e Castro prepared figures and/or tables, authored or reviewed drafts of the paper, approved the final draft.
- José Cola Zanuncio performed the experiments, prepared figures and/or tables, authored or reviewed drafts of the paper, approved the final draft.
- Carlos Frederico Wilcken conceived and designed the experiments, contributed reagents/materials/analysis tools, authored or reviewed drafts of the paper, approved the final draft.

## Data Availability

The raw data are provided in Data S1.

## Supplemental Information

Supplemental information for this article can be found online at http://dx.doi.org/10.7717/peerj.5265#supplemental-information.

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
