# Peer review of "Longevity and survival of Leptocybe invasa (Hymenoptera: Eulophidae), an invasive gall inducer on Eucalyptus, with different diets and temperatures"

_PeerJ, doi:10.7717/peerj.5265_

## Round 0.1 · original submission · Major Revisions

· Academic Editor

Major Revisions

In addition to the excellent comments of the 3 reviewers, I would like to see details of the Kaplan-Meier models presented in the Results.

·

Basic reporting

Survival of one insect pest is experimentally measured under lab conditions.

The paper is clear and well organized, but two part must be changed:
1.the abstract which deals mainly on protocols.
2.the beginning of discussion which should summaries main results.

Hypotheses are in majority supported by results (see section 3 of comments).
An additionnal serie with males would be welcome because males are less resistant than females in most insects.

Experimental design

Good, some details could be added:
1. are females virgin or mated or undefined?
2. temperature bioassays, are females placed individually in vials?
3. were females controlled and observed, for fresh weigth or body size, feeding behavior, general activity, or the day before death...?

Validity of the findings

The question is what are the best conditions for survival of females for a controlled culture of this pest?
The proposed protocol does not answer to the whole question, because 1. groups of females were not tested and it is known that living in groups induces mortality, 2. females are not controlled for egg laying, and it is also known that females which lay eggs leave less times than constrained females (trade off between reproduction and survival). I suggest to reduce the goal of the paper to the first step of mass culture, that is survival of individual non reproductive females.
In the discussion, a parallel is made with nectar (line 149), it may be true, but honey is not nectar, and many published works use honey; I suggest to refer to such data.

line 158, what is C. noackae?

Additional comments

Paper could be improved without major revisions, and focused on the experimental protocol.

Reviewer 2 ·

Basic reporting

In “Longevity and survival of Leptocybe invasa (Hymenoptera, Eulophidae), an invasive gall inducer on Eucalyptus, with different diest and temperatures” the authors evaluate survival and life span of the blue gum chalcid, a pest in eucalypts. In two separate experiments the authors exposed newly-emerged females to different diets at the same temperature and differ temperatures with the same diet. The study is elegant, clear and well done. This manuscript provides good insight on the life history of this eulophid but would benefit from certain clarifications.

Introduction
The introduction needs improvement in the flow. There is information about L. invasa, the focal species (Lines 70-73), then it moves into general information about all insects overall (Lines 73-87), followed again by specifics about L. invasa (Lines 88-91). I suggest the authors consider improving the flow of this section. Start broad with studies and knowledge about research done in longevity and survival in insects, then move into your study species; what is known about it and why the research your doing is important.
Additionally, the introduction would benefit from more detail about the how biological characteristics affect species especially details on the role of temperature and diet affect in other hymenoptera. Giving more detail would let the reader understand the relevance of this study. Please see the attached PDF for more comments

The other sections are structured clearly.

The figures are representative and useful. The Raw data is shared as supplemental data.

The study is self-contained and the analyses of survival are appropriate for the hypotheses.

Experimental design

The research question is well defined and the methods presented are appropriate. There are however a few clarifications needed (please see comments in the attached PDF).

Have the authors done a power analysis to determine whether 25 individuals/tretament (as I assume that is the number of individuals used/treatment) is enough to be able to compare 6-7 treatments?

Statistical analyses for survival are sound.

There is no mention about the analyses done to test the differences for the longevity. Was is just ANOVAs followed by post hoc tukey tests to determine the differences between each of the treatments?

Additionally, it would be beneficial to include information about which statistical program was used.

Validity of the findings

Many species of hymenoptera have been tested for longevity and survival. These are important life history traits to know to develop proper management practices so this stdy is valid and provides good insight on a pest species.

The results section seems straightforward and well written but there are again some clarifications needed.

Line 120. There is no statistical difference between the longevity of L. invasa fed with 50%, 100% 100%+eucalyptus. You could mention that there seems to be some biological difference; at 50% the individuals live longer.

Line 122. I agree that the 50% solution has higher proportion of survivability but 100% and 100%+eucalypt are similar in that they all have some survival until 12 days. This parallels the longevity; 50%, 100% and 100%+eucalypt are not different from each other.

Lines 124-126. It should be mentioned that 14 and 18C are the temperatures with highest longevity, based on your table there is overlap.

The discussion is well written, it nicely roots this study with the literature. This is what the introduction needs. But there are a couple of minor points (please see attached PDF).

Additional comments

This is an interesting study and clearly it was done with care. I hope the authors are continuing with this work. Maybe trying to determine the interaction between temperature and diet of survival and longevity of L. invasa.

Annotated reviews are not available for download in order to protect the identity of reviewers who chose to remain anonymous.

Reviewer 3 ·

Basic reporting

I have made extensive comments on the manuscript itself. Therefore, I comment here briefly.
The language is fine in most places. In some places (see Abstract, Results, Discussion sections), the language can be tightened so that the meaning is clear.
The authors do a good job of citing literature. However, I would suggest re-looking at the background and motivation parts of the manuscript. Reading through the manuscript, I got a feeling that the authors are torn between two motivations (a) understand the biology of the species such that the knowledge is useful in pest control etc. (b) come up with a method of culturing this species in the lab. The writing has to be such that these two motivations do not often clash.

Experimental design

The experiment is well designed. I suggest that more information be given about (a) Ancestry of the population used for the experiment (b) its regular maintenance protocol (c) reason for choosing the particular treatments (d) sample size (e) results.

Validity of the findings

Findings are simple and straight forward, which is very nice. I would request the authors to reconsider their discussion. As it stands, discussion gives a lot of details about results from other studies. However, it is important to discuss your own results with the two stated aims of this study in mind.

Annotated reviews are not available for download in order to protect the identity of reviewers who chose to remain anonymous.

---

## Round 0.2 · accepted · Accept

· Academic Editor

Accept

I could secure only one re-reviewer for your revision. Both the reviewer and I agree that you have adequately addressed the concerns in the original manuscript.

# ·

Basic reporting

changed as suggested

Experimental design

OK

Validity of the findings

OK

Additional comments

Changes I suggested were made by authors.